# 4Real: Towards Photorealistic 4D Scene Generation via Video Diffusion Models

Heng Yu[1,2,*,†]   Chaoyang Wang [1,*]   Peiye Zhuang[1]   Willi Menapace[1]   Aliaksandr Siarohin[1]
Junli Cao[1]   László A Jeni[2]   Sergey Tulyakov[1]   Hsin-Ying Lee[1]
[1]Snap Inc.    [2]Carnegie Mellon University
Project page: https://snap-research.github.io/4Real/

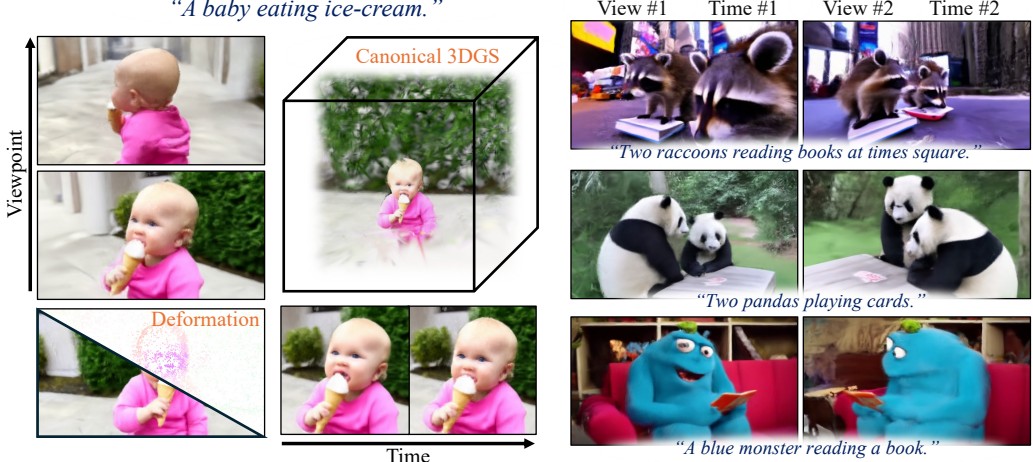

Figure 1: **4Real** is a 4D generation framework that can generate near-*photorealistic* dynamic scenes from text prompts. We use deformable 3D Gaussian Splats (D-3DGS) to model the scene. After generation, We can view the generated dynamic scenes at any timestep from different camera poses.

## Abstract

Existing dynamic scene generation methods mostly rely on distilling knowledge from pre-trained 3D generative models, which are typically fine-tuned on synthetic object datasets. As a result, the generated scenes are often object-centric and lack photorealism. To address these limitations, we introduce a novel pipeline designed for photorealistic text-to-4D scene generation, discarding the dependency on multi-view generative models and instead fully utilizing video generative models trained on diverse real-world datasets. Our method begins by generating a reference video using the video generation model. We then learn the canonical 3D representation of the video using a freeze-time video, delicately generated from the reference video. To handle inconsistencies in the freeze-time video, we jointly learn a per-frame deformation to model these imperfections. We then learn the temporal deformation based on the canonical representation to capture dynamic interactions in the reference video. The pipeline facilitates the generation of dynamic scenes with enhanced photorealism and structural integrity, viewable from multiple perspectives, thereby setting a new standard in 4D scene generation.

---

[*]Equal contribution.
[†]Work done during internship at Snap Inc.

38th Conference on Neural Information Processing Systems (NeurIPS 2024).

# 1 Introduction

As industries ranging from film production to virtual reality seek increasingly immersive and interactive experiences, the ability to generate dynamic 3D scenes over time—essentially, 4D environments—promises to revolutionize how we interact with digital content. Recently, significant advances in image and video generation have been driven by large-scale text-image and text-video datasets, along with the development of diffusion models. Furthermore, image diffusion models have been adapted into 3D-aware multi-view generative models through fine-tuning with limited 3D data, serving as foundational priors for 3D and 4D generation.

In this work, we focus on photorealistic text-to-4D scene generation, Existing 4D generation pipelines, due to the lack of 4D data, typically employ image, multi-view, and video generation models as priors to synthesize 4D samples. However, the multi-view models, which provide critical 3D information, are fine-tuned on static and synthetic 3D assets. As a result, current generated 4D results are predominantly object-centric, lacking photorealism, and limited in their ability to capture complex interactions between objects and environments.

In response, we propose **4Real**, a novel pipeline designed for photorealistic dynamic scenes with dynamic objects within the environment. We discard the reliance on multi-view generative models and exploit video generative models trained on large-scale real-world videos, covering more diverse and general appearance, shape, motion, and the interaction between objects and environments. Compared to existing methods purely relying on score distillation sampling, the proposed pipeline provides more flexible use cases, more diverse results, and requires fewer computations.

The proposed method adopts deformable 3D Gaussian Splats (D-3DGS) as the representation for dynamic scenes. The pipeline unfolds in three steps. First, we begin by creating a reference video featuring a dynamic scene with a pre-trained text-to-video diffusion model. Next, we reconstruct a canonical 3D representation from a selected frame of the reference video. To achieve this, we generate a 'freeze-time' video with camera motion and minimal object movement by applying dataset context embedding and prompt engineering to the video diffusion model. However, practically, the generated video still may contain object motion and is prone to be geometrically inconsistent despite looking temporally smooth. We propose to recognize the inconsistencies as per-frame deformation with respect to the canonical representation and learn these deformations simultaneously with the canonical representation. Finally, we reconstruct temporal deformation from the reference video with the learned canonical representation. We choose a video score distillation sampling (SDS) strategy that renders two types of videos: a fixed-viewpoint video with a static camera, and a freeze-time video with a fixed time step. The fixed-viewpoint video helps learn temporally consistent motions, while the freeze-time video learns multi-view geometrical consistency.

**4Real** achieves text-driven dynamic scene generation with a near-photorealistic appearance and realistic 3D motions. The generated scenes are viewable from different viewing angles and over time, as shown in Figure 1. Our contributions are summarized as follows:

- We propose **4Real**, the first photorealistic text-to-4D scene generation pipeline. Without reliance on biased multi-view image generation models trained with specialized datasets, the proposed pipeline can generate more diverse and near-photorealistic results with dynamic objects within realistic environments.

- We leverage text-to-video diffusion to generate target reference videos and auxiliary freeze-time videos, thereby transforming the generation problem into a reconstruction problem and reducing reliance on time-consuming score distillation sampling steps. We learn from the freeze-time videos canonical 3DGS along with per-frame deformation modeling video inconsistency, and learn from the reference videos temporal deformation.

- The proposed pipeline provides users flexibility in selecting and editing videos that they want to lift to 4D, and can generate high-quality samples in a more reasonable computation budget, taking 1.5 hours on an A100 GPU compared to 10+ hours with competing methods [2, 72].

# 2 Related Works

**Text-to-video generation.** Our pipeline leveraged the advancements in video generation models, which have been spurred by the recent success of text-to-image diffusion models. Some approaches

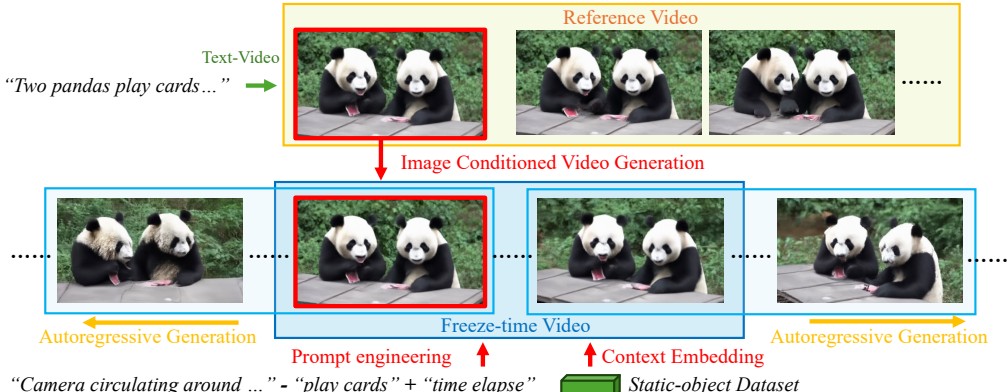

Figure 2: **Generate reference and freeze-time videos.** Given the input text prompt, we first generate a *reference video* using text-to-video diffusion model. The reference video will be our target to perform 4D reconstruction. In order to obtain canonical Gaussian Splats, we generate a *freeze-time video* by performing frame-conditioned video generation with prompt engineering and context embedding. We further perform auto-regressive generation to expand view angle coverage.

extend these capabilities by directly fine-tuning pre-trained image diffusion models [3, 4] or by integrating motion modules while maintaining fixed image models [17, 34, 71]. Others utilize both image and video data using 3D-Uet [20] and cascaded temporal and spatial upsampler [19, 48]. More recently, some efforts [5, 35] propose to replace the Unet architectures with transformer-based architectures [9, 41] for improved quality, scalability, and training efficiency.

**Object-centric 3D and 4D generation.** The challenges of generating 3D and 4D data mostly lie in the scarcity of available data, due to the costly nature of data capture. Conventional 3D generative models [1, 10–12, 22, 28, 38, 50, 57] that train on limited 3D data in various representation fall short in quality and diversity, while 2D diffusion models have shown incredible generalizability using large-scale data. To bridge the gap, Score Distillation Sampling (SDS) [42] and its variants [8, 27, 56, 58, 73] have been developed to utilize pre-trained text-to-image models as priors to generate 3D objects by optimizing parametric spaces for 3D representations. Recently, multiview images have emerged as a popular representation in 3D generation. Multiview image diffusion models [30–32, 47] are fine-uned from image diffusion models using synthetic object dataset [13]. To generate 3D objects, the multiview image diffusion models are used as approximate 3D priors [44, 62], or as providing intermediate medium followed by 3D reconstruction methods [21, 25, 32, 51, 59, 65, 74]. Meanwhile, video diffusion models have also been explored as priors for some early 4D generation attempts [49] or as data generators to provide multiview supervision for 3D model [52]. More recently, image, multiview, and video diffusion models are jointly used as priors for 4D generation [2, 29, 45, 68, 72]. However, they typically produce results biased toward object-centric and non-photorealistic outputs as the 3D priors are obtained from synthetic object datasets.

**4D Reconstruction.** Neural Radiance Fields (NeRFs) [36] has revolutionized 3D scene reconstruction and novel view synthesis. The breakthroughs have been expanded to dynamic scenes. There are two major branches of methods for dynamic NeRFs: one views them as a 4D representation by extending the radiance fields with a time dimension [14, 16, 26, 53, 55, 64], while the other represents a dynamic scene by a canonical space coupled with a deformation field that maps local observation to this space [15, 39, 40, 43, 54, 66]. Recently, 3D Gaussian Splatting (3DGS) [23] has been introduced as a promising representation that facilitates fast neural rendering using explicit Gaussian particles. 3DGS has then been applied to model dynamic scenes with the similar idea of building a deformation field [33, 63, 67, 69]. We adopt deformable 3D Gassian Splats (D-3DGS) as our dynamic scene representation.

## 3 Method

Given input text prompts, we aim to generate photorealistic dynamic scenes including animated objects and detailed backgrounds that exhibit plausible relative scales and realistic interactions with moving objects. We adopt deformable 3D Gaussian Splats (D-3DGS) as our representation. First, we

utilize a text-to-video diffusion model to create a reference video with dynamic scenes. Next, a frame from this reference video serves as the conditional input for the video diffusion model to produce a freeze-time video featuring circular camera motion and minimal object movement. Subsequently, we reconstruct the canonical 3D representation from the freeze-time video. Finally, we reconstruct temporal deformations to align with the object motions in the reference video.

In the following sections, we will first discuss the 4D Gaussian splats representation for dynamic scenes in Section 3.1, the generation of freeze-time videos in Section 3.2, the reconstruction of canonical 3DGS in Section 3.3 and the temporal reconstruction in Section 3.4.

## 3.1 Dynamic Scene Representation

We employ Deformable 3D Gaussian Splats (D-3DGS) to model dynamic scenes, a popular choice for dynamic novel view synthesis tasks due to its low rendering computation cost, state-of-the-art rendering fidelity, and compatibility with existing graphics rendering pipelines [33, 63, 67, 69]. D-3DGS consists of 3D Gaussian splats that represent the static 3D scene at a canonical frame and a deformation field that models dynamic motions.

**Canonical 3D Gaussian splats.** 3D Gaussian Splats [23] consist of a set of 3D Gaussian points, each associated with an opacity value $\alpha$ and RGB color $\mathbf{c} \in \mathbb{R}^3$. The position and shape of each 3D Gaussian point are represented by a Gaussian mean position $\mathbf{x} \in \mathbb{R}^3$ and a covariance matrix, which is factorized into orientation, represented by a quaternion $q \in \mathbb{R}^4$, and scaling, represented by $\mathbf{s} \in \mathbb{R}^3$. 3DGS are rendered by projecting Gaussian points onto the image plane and aggregating pixel values using NeRF-like volumetric rendering equations.

**Deformation field.** To represent motion in 4D scenes, we employ a deformation field $w_{t\text{-deform}}$ that represents the offset of 3DGS from the canonical frame to the frame at time $t$:

$$w_{t\text{-deform}}(\mathbf{x}, t) = (\Delta\mathbf{x}_t, \Delta\mathbf{q}_t). \tag{1}$$

The position and orientation of 3DGS at time $t$ are simply obtained by $\mathbf{x}_t = \mathbf{x} + \Delta\mathbf{x}_t$ and $\mathbf{q}_t = \mathbf{q} + \Delta\mathbf{q}_t$. We implement $w$ using a multi-layer perceptron (MLP), which serves as a general representation for arbitrary deformations. It is worth noting that this can be substituted by more specialized representations, such as linear blend skinning, which is more suitable for articulated motions. In this work, we prefer MLPs due to their simplicity and generality.

## 3.2 Generating Reference and Freeze-time Videos

Given a text prompt, we generate a reference video using video diffusion models. First, we would like to obtain the canonical 3D Gaussian splats for this reference video by creating a freeze-time video conditioned on a selected frame. This process requires a video generation model that can: 1) perform image-to-video generation, ensuring the output video frames remain consistent with the input image, and 2) generate videos with substantial camera motion while maintaining a relatively static scene.

We evaluated several recent video models accessible to the authors, including SVD [3], SV3D [52], VideoCrafter [7], and Snap Video Model [35]. SVD and VideoCrafter support image-to-video generation but have limited camera movement and introduce random motions in non-rigid objects. SV3D creates object-centric 360° videos with a white background, unsuitable for the scene-level generation. The Snap Video Model aligns most closely with our requirements, offering the required capabilities for our application.

**Frames-conditioned pixel-space video diffusion.** The Snap Video Model is a text-to-video diffusion model that diffuses pixel values instead of latents, resulting in less motion blur and fewer gridding effects, which are more common in latent diffusion models. It also tends to produce larger camera and object motions [35]. We use a variant trained to generate videos from both text prompts and an arbitrary set of video frame inputs. This feature enhances flexibility for autoregressive generation, particularly when creating freeze-time videos with extensive view coverage. Details of training and inference are in the Supplementary.

**Context conditioning and prompt engineering.** The Snap Video Model is trained on mixed video datasets, with a unique context embedding for each dataset to guide the model in generating videos that follow the specific distribution. One dataset consists of static real-world objects with circular camera trajectories, helping the model generate freeze-time videos with large camera motions and

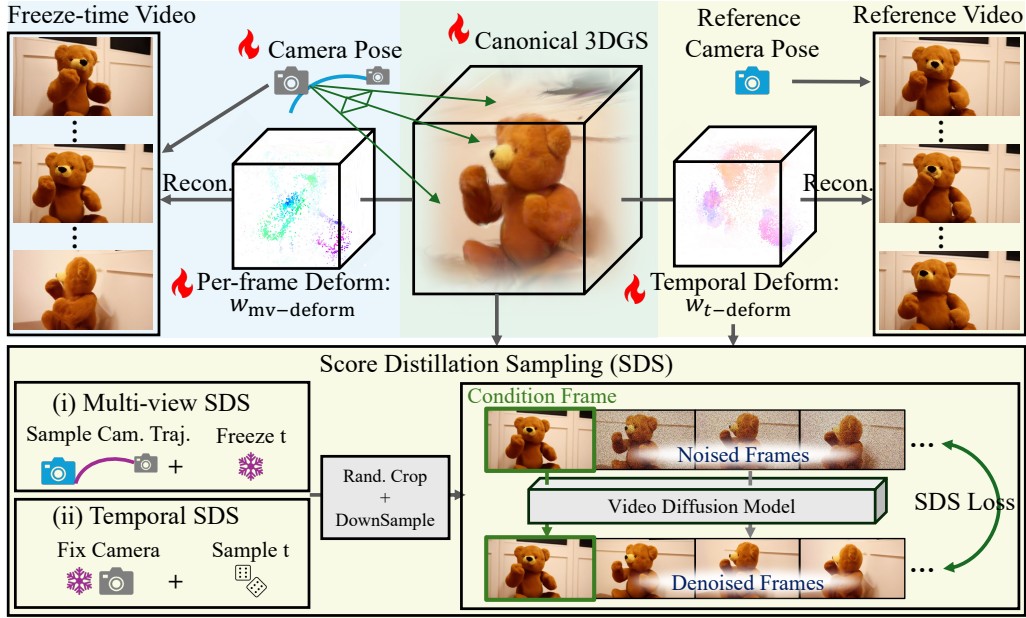

Figure 3: **Reconstructing Deformable 3DGS.** First, we reconstruct the canonical 3DGS using the freeze-time video. As the generated freeze-time video may still contain object motions and geometric inconsistencies, we propose to model these imperfections as per-frame deformations that are learned jointly with the canonical 3DGS. Then, we reconstruct the temporal deformation from the reference video with the learned canonical 3DGS. In addition to the reconstruction loss and motion regularization, we propose a video SDS strategy. The video SDS strategy includes a multi-view SDS, using videos of freeze time and sampled camera trajectory, and a temporal SDS, using videos of fixed camera and sampled time steps. 🔥 denotes learnable parameters.

minimal object movements when its context embedding is used. The success rate can be further improved by adding phrases such as "Camera circulating around" and "time elapse" to the beginning and end of the input prompt, while omitting action verbs. Despite this, limited by the current capability of the Snap Video Model, multi-view inconsistency and moderate object motion may still occur, which our reconstruction steps in Section 3.3 address.

**Extending view coverage.** The raw output from the Snap Video Model is a sequence of 16 frames, resulting in limited view coverage. To extend coverage, we perform iterative autoregressive generation. At each step, we condition on the 8 previously generated frames, guiding the model to generate 8 new frames that follow the camera trajectory of the prior frames. We extend the view coverage in both forward and backward directions until it reaches a sufficient amount, such as $180°$. Covering $360°$ is currently challenging due to background clutter and the model's limitation in generating loop videos.

### 3.3 Robust Reconstruction of 3DGS from Noisy Freeze-time Videos

Constrained by the capability of current video models, generated freeze-time videos may still contain multi-view inconsistencies, preventing direct reconstruction of high-quality static 3DGS. These inconsistencies can be: 1) temporally consistent but geometrically incorrect 2D video, and 2) videos with object motions. To address these imperfections, we propose the following treatments.

**Representing multi-view inconsistency as deformation.** We treat multi-view inconsistency as per-frame deformation with respect to the canonical 3DGS. Specifically, we introduce another deformation field $w_{\text{multi-view}}(\mathbf{x}, k) = (\Delta\mathbf{x}_k, \Delta\mathbf{q}_k)$ similarly defined as in Eq. (1), which outputs changes in position and orientation of a Gaussian point at the canonical position $\mathbf{x}$ with respect to its new position and orientation at the $k$-th frame. We choose the canonical frame to be the one from the reference video that is used as the conditional input to generate the freeze-time video. This implies zero-valued outputs for the deformation field at the canonical time frame.

We jointly optimize the deformation field $w_{\text{multi-view}}$ and the parameters for the 3DGS by minimizing a combination of the following losses.

**Image reconstruction loss** computes the difference between images $\mathcal{I}_{\text{GS}}$ rendered by the deformed 3DGS and the frames $\mathcal{I}_k$ from the freeze-time video.

$$\mathcal{L}_{\text{recon}} = \sum_k \|\mathcal{I}_{\text{GS}}(\mathbf{x} + \Delta\mathbf{x}_k, \mathbf{q} + \Delta\mathbf{q}_k, \mathbf{s}, \alpha, \mathbf{c}, \mathbf{P}_k) - \mathcal{I}_k\|_1, \tag{2}$$

where $k$ denotes the frame index, $\mathbf{P}_k$ represents the camera projection matrix which is initialized using Colmap [46], then jointly finetuned with the 3DGS, and with some abuse of notations, $(\mathbf{x} + \Delta\mathbf{x}_k, \mathbf{q} + \Delta\mathbf{q}_k, \mathbf{s}, \alpha, \mathbf{c})$ denotes parameters for the deformed 3DGS.

**Small motion loss.** Minimizing $\mathcal{L}_{\text{recon}}$ alone is insufficient to obtain plausible 3DGS, since there can be infinitely many pairs of deformation fields and 3DGS that produce identical image renderings. To address this under-constrained problem, we introduce the assumption that the deformations should be as small as possible. Specifically:

$$\mathcal{L}_{\text{small mo.}} = \sum_k \|\Delta\mathbf{x}_k\|_1 + \|\Delta\mathbf{q}_k\|_1. \tag{3}$$

**Score distillation sampling (SDS) loss.** Hand-crafted heuristics like $\mathcal{L}_{\text{small mo.}}$ may be insufficient for plausible canonical 3DGS, especially with significant multi-view inconsistencies in the freeze-time videos. For improved regularization, in the *later* stage of optimization, we employ score distillation sampling. This method ensures that the rendering of canonical 3DGS on randomly sampled camera trajectories aligns with the distribution dictated by the video diffusion model.

During each optimization step, we sample a camera trajectory by first randomly selecting a camera pose $\mathbf{P}_k$ from the camera trajectories of the freeze-time video and then randomly perturbing it to create $\hat{\mathbf{P}}$. The trajectory $\{\hat{\mathbf{P}}_{k=1,...,16}\}$ is uniformly interpolated between the canonical camera pose and $\hat{\mathbf{P}}$. We then render the canonical 3DGS using this sampled trajectory to generate video frames:

$$\mathcal{I}_{\text{can-GS}}^k := \mathcal{I}_{\text{GS}}(\mathbf{x}, \mathbf{q}, \mathbf{s}, \alpha, \mathbf{c}, \hat{\mathbf{P}}_k), \quad \forall k = 1, ..., 16. \tag{4}$$

To perform score distillation sampling, we add Gaussian noise to the rendered images $\mathcal{I}_{\text{can-GS}}^k$ and then use the video diffusion model to produce one-step denoised images. Since the sampled camera trajectory originates from the canonical camera pose, the first frame $\mathcal{I}_{\text{can-GS}}^0$ of the rendered videos is identical to the corresponding frame $\mathcal{I}_c$ from the freeze-time video, allowing us to substitute $\mathcal{I}_{\text{can-GS}}^0$ with $\mathcal{I}_c$, which can then serve as the conditional frame for the diffusion model. More specifically:

$$\{\hat{\mathcal{I}}_2, \hat{\mathcal{I}}_3, ..., \hat{\mathcal{I}}_{16}\} = f_{\text{denoise}}(\mathcal{I}_{\text{can-GS}}^2 + \epsilon_2, \mathcal{I}_{\text{can-GS}}^3 + \epsilon_3, ..., \mathcal{I}_{\text{can-GS}}^{16} + \epsilon_{16}|\mathcal{I}_c, \sigma), \tag{5}$$

where $\epsilon$'s represent the noise applied to the video frames, $\sigma$ denotes the noise level for the current diffusion step, and $\hat{\mathcal{I}}_k$ indicates the $x_0$ prediction for each frame. Using $\mathcal{I}_c$ as the image condition input is crucial because it allows the diffusion model to provide more precise supervisions, and sharper details that are aligned with the reference video.

Then SDS is implemented by minimizing the sum-of-square difference between the rendered video frames $\mathcal{I}_{\text{can-GS}}^k$ and the denoised frames $\hat{\mathcal{I}}_k$, as follows,

$$\mathcal{L}_{\text{SDS}} = \|\mathcal{I}_{\text{can-GS}}^k - \lceil\hat{\mathcal{I}}_k\rceil\|^2. \tag{6}$$

In this context, we use $\lceil \cdot \rceil$ to indicate that the variables are detached from the automatic differentiation computation graph. It is straightforward to show that the gradient of $\mathcal{L}_{\text{SDS}}$ is proportional to the original gradient formulation by Poole *et al.* [42].

**Sampling with low-res pixel-space diffusion model.** The video diffusion model we employ is a two-stage pixel-space model. It first generates a video at a low resolution of $36\times64$, which is then upsampled to a resolution of $288\times512$. For SDS, we employ only the first-stage low-resolution model, as it is significantly less computationally demanding and not straightforward to apply both stages simultaneously. Fortunately, utilizing just the low-resolution SDS is adequate to yield high-quality results. This effectiveness stems from our method relying more on reconstruction loss to enhance fine details, rather than depending primarily on SDS.

To compute the SDS loss, we first downsample the rendered video frames to low resolution. Before the downsampling, frames are randomly shifted and cropped to mitigate aliasing effects.

**Remarks on computational efficiency.** Previous methods employing SDS are typically computationally intensive, making them less practical for real-world applications. For instance, executing one step of SDS training using single precision on A100-80G GPUs with SVD can take $\approx 2$ seconds, not including the time required for rendering. Moreover, GPU memory often becomes overloaded with SVD, which cannot compute gradients for more than four frames simultaneously due to substantial memory demands for backpropagation through the image encoder.

In contrast, our method implements SDS using the low-resolution, pixel-space Snap Video Model. This model leverages a compressed transformer architecture [9], which is significantly faster, requiring $\approx 200$ ms per step. It is also more memory-efficient, allowing for gradient computation across all 16 frames in a single batch. This efficiency makes our approach more viable for practical applications.

### 3.4 Reconstruct Temporal Deformation

After obtaining canonical 3DGS representation, we proceed to generate temporal motion by fitting the deformation field (as detailed in Eq. (1)) to align with the reference video.

**Image alignment loss** computes the similarity between the rendered frames $\mathcal{I}_{GS}^t$ and the reference video frames $\mathcal{I}_t$, combining pixel-wise intensity loss and structural similarity index measure (SSIM) loss [60], *i.e.*, $\mathcal{L}_{align} = \sum_t \|\mathcal{I}_{GS}^t - \mathcal{I}_t\| + \text{SSIM}(\mathcal{I}_{GS}^t, \mathcal{I}_t)$.

**Deformation regularization.** To ensure that the deformation field learns plausible motions, we incorporate losses that enforce both spatial and temporal smoothness in the motion trajectories of each Gaussian splat, following CoGS [69] and Dynamic 3DGS [33]. These as-rigid-as-possible losses encourage minimal variation in both translational and rotational motions of Gaussian splats within their spatial and temporal neighborhoods. We refer readers to the respective papers for detailed treatment. Implementation details specific to our adaptation are included in the supplementary.

**Joint temporal and multi-view SDS.** Relying solely on hand-crafted regularization often leads to noticeable artifacts in novel view rendering, especially when the camera position deviates significantly from that in the reference video. Therefore, in the later stages of optimization, we employ SDS to ensure that the rendered video frames adhere to the distributions of real videos. For evaluating the SDS loss, we render two types of videos:

1. *Fixed-Viewpoint Videos:* The camera remains static while we sample time steps to deform the 3D Gaussian Splats. To ensure that the first frame can be used as a conditional input for the video diffusion model, the sampled time steps always start from 0. This initial frame is rendered using the canonical 3D Gaussian Splats without any deformation.

2. *Freeze-Time Videos:* With a fixed time step, we sample a continuous sequence of camera poses starting from the reference video's camera pose. This setup allows us to directly use the corresponding frame from the reference video as the conditional input to the video diffusion model.

Applying SDS loss to both fixed-viewpoint and freeze-time videos is crucial. The former regularizes D-3DGS for *temporally* consistent motions, and the latter ensures *multi-view* consistency, aligning D-3DGS motions with the reference video and avoiding multi-perspective optical illusions.

**Improving details by re-upsampling** We optionally perform an additional step to enhance the rendering details further. We found that downsampling and then re-upsampling the videos rendered by Gaussian splatting results in fewer artifacts and higher fidelity. Therefore, we render a video with continuously changing viewpoints and time, and then re-upsample it using the upsampler of the video diffusion model. The resulting video serves as a reconstruction target to fine-tune the Gaussian splats, thereby improving the overall quality.

## 4 Experiment

We evaluated our method using a set of 30 text prompts sourced from related works [2, 35, 72]. Please refer to the included webpage for the complete results. As illustrated in Figure 4, our method successfully generates scenes featuring complex illumination and semi-transparent objects such as water. Additionally, it demonstrates flexibility in creating diverse multi-object scenes.

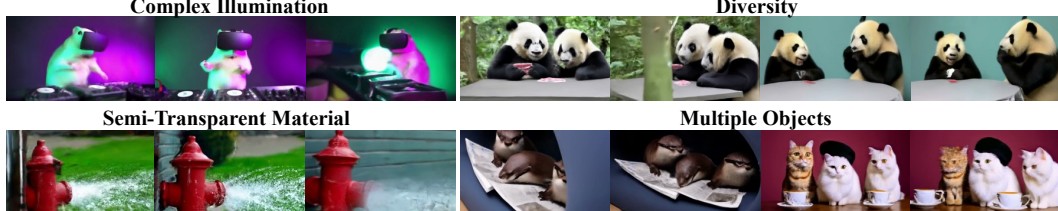

Figure 4: **Generating challenging scenes. 4Real** is able to generate dynamic scenes with complex illumination and (semi)-transparent materials such as water. It is flexible to produce diverse content and generate multi-object scenes. Please refer to our supplementary webpage for the complete results.

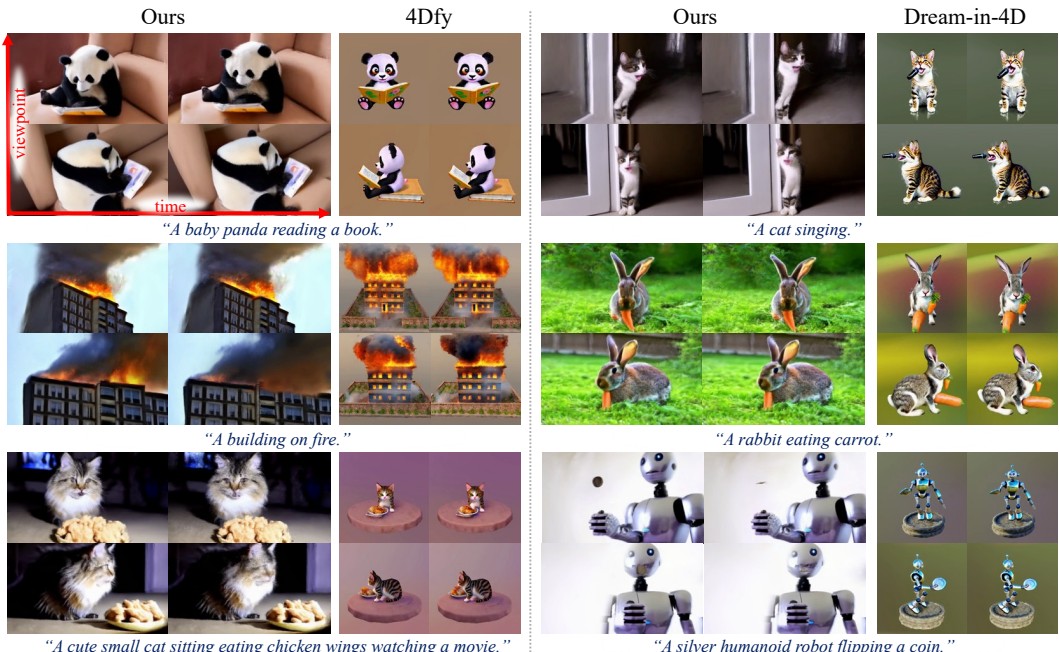

Figure 5: **Qualitative comparison** to state-of-the-arts *object-centric* 4D generation methods.

**Implementation details.** The optimization consists of two stages: first, reconstructing the canonical 3DGS, and then learning the temporal deformation. At each stage, we perform SDS only during the last 5k iterations, and we anneal the diffusion time steps similarly to the scheduler used in Hifa [73]. Throughout our experiments, we run 20k iterations per stage, totaling 1.5 hours on a single A100 GPU. This duration is significantly shorter than that required by recent state-of-the-art methods, which typically require over 10 hours [2, 72]. Details are included in the Supplementary.

**Compare to object-centric text-4D generation.** To the best of our knowledge, there are currently no existing text-to-4D scene generation methods, so we choose to evaluate our method against recent state-of-the-art object-centric text-to-4D generation methods. Specifically, we conduct a user study against 4Dfy [2] and Dream-in-4D [72], which are based on NeRF representation. These methods are known to achieve some of the best visual quality compared to other methods [29, 45, 49] using the same representations.

The user study was conducted using Amazon Turk, involving 30 evaluators per video pair. In each session, evaluators were presented with two anonymized videos. Each video depicted a dynamic object or scene, with the camera moving along a circular trajectory and stopping at four fixed poses to highlight object motions. We obtained 16 videos for 4Dfy and 14 videos for Dream-in-4D from their respective project web pages. Evaluators were tasked with selecting their preferences based on seven criteria: *motion realism*, *foreground/background photo-realism*, *3D shape realism*, *general realism*, *significance of motion*, and *video-text alignment*. As shown in Figure 6, our method outperformed the competition in every category. Sample frames are displayed in Figure 5, and all videos used in the study are available in the supplementary materials.

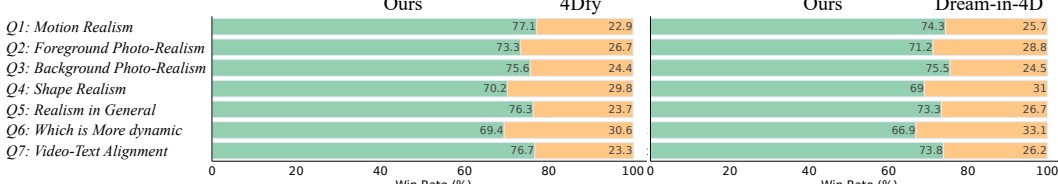

Figure 6: **Comparative user study** with state-of-the-art *object-centric* 4D generation methods.

|  | X-CLIP ↑ | Visual Quality ↑ | Temporal Consistency↑ | Dynamic Degree ↑ | T-V Alignment ↑ | Factual Consistency ↑ |
|---|---|---|---|---|---|---|
| 4Dfy [2] | 20.03 | 1.43 | 1.49 | 3.05 | 2.26 | 1.30 |
| Ours | **24.23** | **2.43** | **2.17** | **3.15** | **2.91** | **2.49** |
| Dream-in-4D [72] | 19.52 | 1.34 | 1.37 | 3.02 | 2.27 | 1.20 |
| Ours | **24.77** | **2.41** | **2.15** | **3.14** | **2.89** | **2.46** |
| AYG [29] | 19.87 | **2.49** | 2.09 | 3.15 | 2.80 | 2.47 |
| Ours | **23.09** | 2.44 | **2.16** | **3.16** | **2.90** | **2.50** |

Table 1: **Quantitative comparison against baselines** using X-CLIP [37] and VideoScore [18].

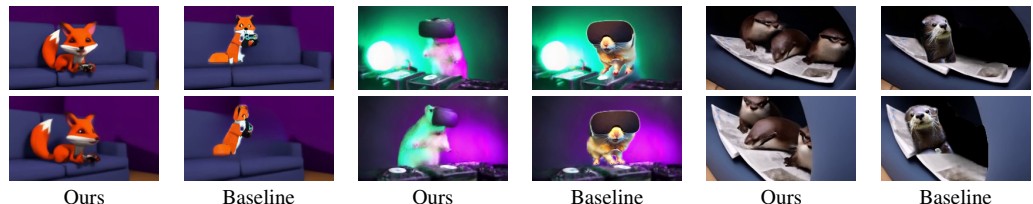

| Ours | Baseline | Ours | Baseline | Ours | Baseline |

Figure 7: **Qualitative comparison against baseline of 3D scene generation + 4D object-centric generation.** Our result is more natural in terms of object placement, motion and lighting.

In addition to existing methods, we also implement and compare with a straightforward baseline: combining 3D scene generation with 4D object-centric generation. Although straightforward, implementation with high quality is challenging due to: (1) Inserting 3D objects with realistic and physically correct placement is not trivial. Common issues include floating objects, misaligned scales between objects and background, and unrealistic scene layout. (2) Generating object motions that align well with the backgrounds is difficult, such as ensuring a generated target subject sits on a sofa rather than stands on it. (3) Complex procedures are needed to relight objects to match environment lighting. We tried our best to create a few baseline results with heavy manual efforts, as shown in Figure 7. We create the 3D background by removing the object from our generated freeze-time video. We then employ 4Dfy [2] to generate the foreground objects. Finally, we manually insert the objects into the background with the most plausible position and scale we can find. Despite these efforts, we find the inserted object appears disconnected from the background, especially when compared to our results.

On top of user study, we also provide quantitative results in Table 1 using employ the X-CLIP score [37], a minimal extension of the CLIP score for videos. We also run VideoScore [18], a video quality evaluation model trained using human feedback, which provides five scores assessing visual quality, temporal consistency, text-video alignment, and factual consistency. Our method significantly outperformed other methods in these metrics. However, we remain conservative about the effectiveness of these metrics since they have not been thoroughly studied and evaluated yet.

**Ablation studies.** We diagnose our system by removing individual components from the pipeline.

*Per-frame deformation* is crucial for reconstructing high-quality canonical 3D representations from noisy freeze-time videos. As illustrated in Figure 8, the absence of per-frame deformation results in significant artifacts, such as on the faces of the panda and the cat, and in blurrier areas like the boundaries of doors in a kitchen.

*Small motion regularization* is employed in our pipeline. While view-dependent deformation can improve underfitting, it can also lead to overfitting to the reference video, resulting in noisy reconstruction. We thus adopt a regularization loss of the magnitude of the deformation to mitigate the issues. As shown in Figure 9.

| Per-frame Deform | Multi-view SDS | Freeze-time Video | Joint Temp. & MV SDS |
|---|---|---|---|
| w/o   with | w/o   with | w/o   with | w/o   with |

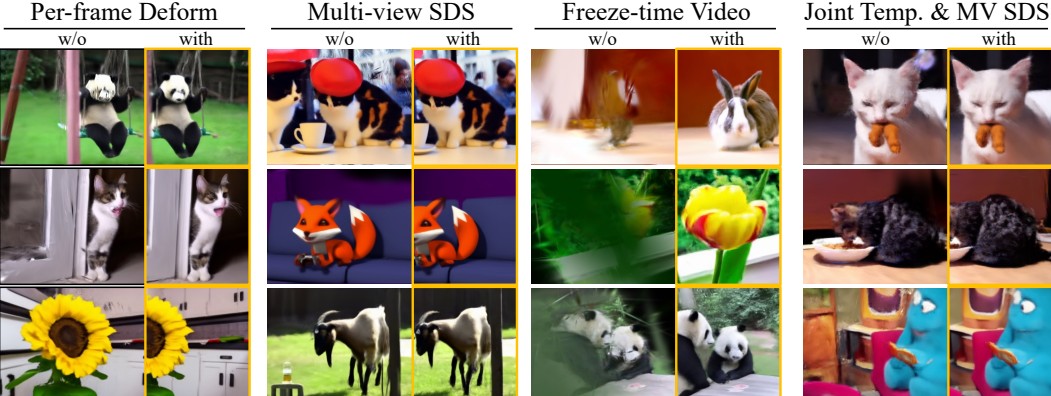

Figure 8: **Ablation study** of the impact of removing each component from the proposed pipeline.

w/o deform   w/o reg      full model         w/o deform   w/o reg      full model

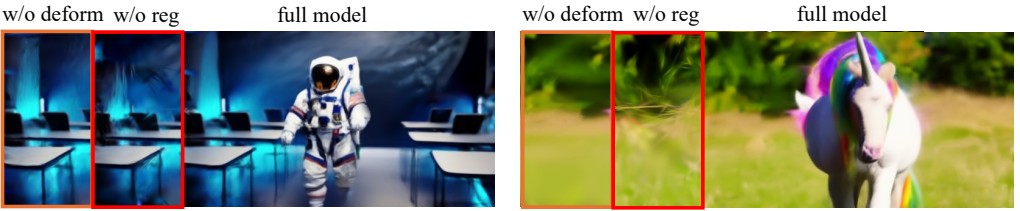

Figure 9: **Ablation on per-view deformation and small motion regularization.** Removing per-view deformation results in underfitting the regions which are not geometrically consistent. Removing small motion regularization of the per-view deformation field causes overfitting to the freeze-time field, leading to noisy results.

*Multi-view SDS* aids in noise reduction and enhances shape realism. For example, as illustrated in Figure 8, it improves the depiction of the human head in the background, the upper boundary of the sofa, and the limbs of the goat.

*Freeze-time video:* Removing the step of generating freeze-time videos and the corresponding reconstruction loss $\mathcal{L}_{recon}$ will result in systematic failure. This is partly because we apply SDS only in the later stages of training, which is insufficient to correct significant reconstruction errors.

*Joint multi-view and temporal SDS* helps improve the sharpness of the final result during the reconstruction of temporal deformation. It also helps prevent erroneous artifacts, such as the floating spot on the edge of the cat's ear and the dark shadows above the cat's back, as shown in Figure 8.

We have included additional samples from the ablation study in the supplementary web page.

## 5   Discussion and Conclusion

We propose **4Real**, which, to our knowledge, is the first method capable of generating near-photorealistic 4D scenes from text inputs.

However, our method has **limitations**: 1) It inherits limitations from the underlying video generation model, such as video resolution, blurriness or artifacts during fast motion, particularly with thin or small objects, and occasional text-video misalignment. 2) Reconstructing from videos with dynamic content is challenging, and our method may fail due to inaccuracies in camera pose estimation, rapid movements, the sudden appearance and disappearance of objects, and abrupt lighting changes. 3) Our method does not produce high-quality geometry such as meshes, due to the limitations of using 3DGS. 4) It still takes over an hour to generate a 2-second 4D scene.

For **future works**, we envision the availability of stronger video generation models with more accurate camera pose and object motion control [6, 61], incorporating cross-frame attention [70] when generating freeze-time videos and utilizing feedforward 3D reconstruction [51] would help address these limitations.

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

# A    Masked Snap Video Model Details

We employ a masked variant of the Snap Video model that is trained to generate videos using both text and a set of known frames as conditioning. Given a set of known frames at arbitrary positions in the input sequence, the masked Snap Video model produces an output video whose frames at those positions match the known frames. This mechanism allows the model to perform various tasks such as image animation, frame interpolation and temporal extension of videos by providing known frames in the corresponding arrangements.

Tha masked model variant is obtained by extending the original Snap Video model's input channels by 3 to accomodate the conditioning frames input, and by training the model for an additional 200k steps using a masked video modeling objective, while keeping the other training parameters unchanged.

In particular, we adopt a set of masking strategies which define the set of known frames provided as conditioning during training, each of which is randomly applied to an input batch element with probability $m$:

- *Unconditional generation* $m = 0.6$. Mask the whole input video. This setting corresponds to the original Snap Video setup, where no known frame is present.
- *Bernoulli* $m = 0.3$. Mask each frames of the input video with probability $(1 - 1/16)$, leaving on average a single frame unmasked.
- *Frame interpolation* $m = 0.075$. Mask all video frames apart from a set of frames sampled at regular time intervals.
- *Video extension* $m = 0.075$. Mask the last N video frames.

# B    Deformation Regularization Details

Our motion deformation network predicts position offsets ($\Delta \mathbf{x}$), rotation offsets ($\Delta \mathbf{q}$), and scale offsets ($\Delta \mathbf{s}$) for each time step. To prevent unnecessary movements (e.g., in the background) and to ensure movement consistency, we apply an $L_1$ norm regularization as follows:

$$\mathcal{L}_{\text{norm}} = \frac{1}{N} \sum_{i=1}^{N} (\|\Delta \mathbf{x_i}\| + \|\Delta \mathbf{q_i}\| + \|\Delta \mathbf{s_i}\|). \tag{7}$$

We introduce a difference loss, denoted as $\mathcal{L}_{\text{diff}}$, to ensure that the movement or trajectory of each Gaussian is consistent and smooth over time. This loss penalizes abrupt changes in the trajectory, promoting smoother transitions. The difference loss is formulated as follows:

$$\mathcal{L}_{\text{diff}} = \frac{1}{N} \sum_{i=1}^{N} \|\Delta \mathbf{x_{i,t}} - \Delta \mathbf{x_{i,t-1}}\|. \tag{8}$$

Here, $\Delta \mathbf{x_{i,t}}$ represents the position offset of Gaussian $i$ at time $t$. We also borrow the local-rigidity loss $\mathcal{L}_{\text{rigid}}$ and rotational loss $\mathcal{L}_{\text{rot}}$ from [33] (For a more comprehensive understanding, please refer to their detailed paper). For the local-rigidity loss, a weighting scheme is applied that utilizes an unnormalized Gaussian weighting factor. This factor assigns different weights to each point based on its proximity to the center of the Gaussian distribution. The unnormalized Gaussian weighting factor is defined as follows:

$$w_{i,j} = \exp\left(-\lambda_w \|\mathbf{x}_{j,0} - \mathbf{x}_{i,0}\|_2^2\right) \tag{9}$$

Using this weighting scheme, a local-rigidity loss is employed, denoted as $\mathcal{L}^{\text{rigid}}$, defined as follows:

$$\mathcal{L}_{\text{rigid}}^{i,j} = w_{i,j} \|(\mathbf{x}_{j,t-1} - \mathbf{x}_{i,t-1}) - \mathbf{R}_{i,t-1} \mathbf{R}_{i,t}^{-1} (\mathbf{x}_{j,t} - \mathbf{x}_{i,t}) \|_2, \tag{10}$$

$$\mathcal{L}_{\text{rigid}} = \frac{1}{k|G|} \sum_{i \in G} \sum_{j \in \text{knn}_{i;k}} \mathcal{L}_{\text{rigid}}^{i,j}. \tag{11}$$

This loss ensures that for each Gaussian $i$, neighboring Gaussians $j$ move in a manner consistent with the rigid body transformation of the coordinate system over time. Specifically, this means that the relative positions and orientations of neighboring Gaussians are maintained as the coordinate system evolves, ensuring a coherent and physically plausible motion pattern.

Additionally, we incorporate a rotational loss $\mathcal{L}^{\text{rot}}$ to ensure consistent rotations among neighboring Gaussians across different time steps. This loss function helps maintain the relative orientations of nearby Gaussians as they evolve, preserving the overall rotational coherence of the system. The rotational loss is expressed as:

$$\mathcal{L}_{\text{rot}} = \frac{1}{k|G|} \sum_{i \in G} \sum_{j \in \text{knn}_{i;k}} w_{i,j} \|\hat{\mathbf{q}}_{j,t} \hat{\mathbf{q}}_{j,t-1}^{-1} - \hat{\mathbf{q}}_{i,t} \hat{\mathbf{q}}_{i,t-1}^{-1}\|_2, \tag{12}$$

Here, $\hat{\mathbf{q}}$ represents the normalized quaternion rotation of each Gaussian. The $k$-nearest neighbors, identified in the same manner as in the preceding losses, are utilized to ensure consistent rotational behavior among neighboring Gaussians.

## C   Stage-wise Training

Our method consists of two main components: canonical space reconstruction and motion fitting.

1. **Canonical Space Reconstruction**

   (a) **Warm-up Stage:**
   - Train the network for 3,000 iterations without any per-frame deformations to establish a baseline for the canonical space. This stage follows the same method as described in [23].

   (b) **Learning Deformations and GS Optimization:**
   - Learn deformations for each frame and optimize the Gaussians over the course of 20,000 iterations.
   - Implement GS growth (including splitting, cloning, and pruning) using the strategy described in [23] during the first 15,000 iterations.

   (c) **Multi-view SDS:**
   - Enable multi-view SDS in this stage from 15,000 iterations to help learn a good canonical space.

2. **Motion Fitting**

   (a) **Initial Motion Fitting:**
   - Fit the motion over the next 10,000 iterations using only reference views at different timesteps.

   *Note: Cease GS optimization to focus solely on learning motion deformations.*

   (b) **GS Growth for Motion Compensation:**
   - Resume GS growth using the same strategy for an additional 5,000 iterations to address missing points in the canonical space caused by motion.

   *Note: Fix the learned canonical space from the previous step, and perform GS growth by focusing solely on the newly added Gaussians.*

   (c) **Simultaneous Fine-Tuning:**
   - Fine-tune both the GS and the motion deformation network simultaneously using all training frames consisting of different camera views and timesteps.

   (d) **Joint Temporal and Multi-view SDS:**
   - Enable temporal and multi-view SDS jointly in this stage when doing fine-tuning from 35,000 iterations and last for 5,000 iterations to help learn a consistent motion.

# D  Implementation Details

## D.1  Multi-view and Temporal SDS

During training with SDS, we alternately perform multi-view and temporal SDS. For each iteration, we randomly sample either a camera pose or a timestep. This sampling determines the specific spatial (view) dimension or the temporal (time) dimension to train. By alternating between these two types of SDS, we ensure comprehensive coverage of both spatial and temporal aspects. This strategy enhances the overall consistency and robustness of the model. For each SDS step, we randomly sample $M = 16$ frame poses and perform rendering, either following the sequence of camera movement or based on the order of time. It is important to note that we perform noisy virtual trajectory sampling during multi-view SDS. Specifically, we randomly sample a camera pose that is different from the reference pose used to generate the temporal video. Subsequently, we sample a virtual camera pose within a specified radius of a circle centered on the sampled pose. By interpolating between this virtual camera pose and the reference camera pose, we generate $M = 16$ frame poses for rendering. This approach ensures a diverse set of viewpoints, enhancing the robustness and accuracy of the multi-view SDS process. When inputting our high-resolution rendered images into the low-resolution model for SDS, we first apply one of two preprocessing steps with equal probability. The image is either randomly shifted by $S - 1$ pixels (where $S$ is the downsampling scale) or randomly cropped to half its size. Following this, we downsample the image and input it into the low-resolution model. This preprocessing ensures that the low-resolution model receives varied inputs, enhancing the robustness of the training process.

## D.2  GS Growth for Motion Fitting

During the GS growth for motion fitting, we fix the attributes (position, rotation, scale, opacity and color) of the learned canonical Gaussians by stopping the gradient flow to them. Instead, we focus on performing splitting, cloning, and pruning operations based on the gradients of the newly added Gaussians. These new Gaussians are introduced at the beginning of the growth stage, specifically according to the gradients derived from the canonical Gaussians. This method ensures that the canonical structure remains stable while allowing the newly added Gaussians to adapt and optimize the motion fitting process.

## D.3  Deformation Network

For the training process, we adopted the differential Gaussian rasterization technique from 3D GS [23] and utilized the same deformation network structure as described in [67] for both per-frame deformation and motion deformation. Both deformation networks consist of an MLP $\mathcal{F}_\theta$: $(\gamma(\mathbf{x}), \gamma(t)) \to (\Delta\mathbf{x}, \Delta\mathbf{q}, \Delta\mathbf{s})$, which applies position embedding $\gamma$ on each coordinate of the 3D Gaussians and time (or per-frame deformation code) and maps them to their corresponding deviations in position, rotation, and scaling. The weights $\theta$ of the MLP are optimized during this mapping process. Each MLP $\mathcal{F}_\theta$ processes the input through eight fully connected layers, each employing ReLU activations and containing 256-dimensional hidden units, resulting in a 256-dimensional feature vector. This feature vector is then passed through three additional fully connected layers (without activation functions) to independently output the offsets for position, rotation, and scaling over time. It is worth noting that, similar to NeRF, the feature vector and the input are concatenated at the fourth layer, enhancing the network's ability to capture complex deformations.

## D.4  Training Setup

The initialization of the Gaussians was based on Structure-from-Motion (SfM) results obtained from COLMAP [46]. We adhered to the training settings specified in [23] for the canonical GS training. The learning rate for the positions of the Gaussians was set to undergo exponential decay, starting from $1.6 \times 10^{-4}$ and decreasing to $1.6 \times 10^{-6}$. Additionally, the learning rate for both scale and rotation parameters was maintained at $1 \times 10^{-3}$ throughout the training process. The learning rate for the deformation network was set to exponentially decay from $1 \times 10^{-3}$ to $1 \times 10^{-5}$. The optimization across these processes was conducted using the Adam optimizer [24] with $\beta$ parameters set to $(0.9, 0.999)$. All experiments were performed on single NVIDIA A100 GPUs with 80GB of memory.

# E  Key Hyperparameters

## E.1  Loss Weighting

For the regularization terms mentioned above, we assign a weight of 0.01 to each within the overall loss function, ensuring they contribute appropriately to the optimization process. For the reconstruction loss, we use a weight of 1. Additionally, for the SDS loss, we differentiate the weights based on the type of SDS being applied: a weight of 20 for temporal SDS and a weight of 5 for multi-view SDS. These weights are chosen to effectively capture motion dynamics and stabilize the training process.

## E.2  GS Growth Threshold

For the GS growth stage during the canonical space reconstruction, we set the opacity threshold to $\tau_\alpha = 5 \times 10^{-3}$ and the gradient threshold to $\tau_{grad} = 2 \times 10^{-4}$. These thresholds help control the addition of new Gaussians based on their opacity and gradient values. In contrast, for the GS growth stage during motion fitting, we use a relatively higher opacity threshold of $\tau_\alpha = 1 \times 10^{-2}$ to avoid introducing redundant Gaussians. This higher threshold ensures that only significant Gaussians are added, which aids in maintaining efficiency and relevance during the motion fitting process.

# F  User Study Details

The user study was conducted through a single survey comprising seven questions. Each question asked the evaluators to compare two videos: one rendered with our method and the other with a different method using the same text prompt as input. Evaluators indicated their preferred video based on various aspects, as illustrated in Fig. 10. The evaluators were given the following instructions for each metric.

- **Motion Realism:** Take into consideration of magnitude, smoothness, consistency of the motion. Pay close attention to deformed limbs of human and animals and unnatural deformations.
- **Foreground Photo-realism:** Assess realism of the appearance of the foreground object, do not consider the motion factor. That is the main object that looks more like taken with real video camera, pay attention to unnatural color or styles of the objects.
- **Background Photo-realism:** Assess realism of the appearance of the background. That is the background that looks more like taken with real video camera, pay attention to unnatural color or styles of the objects or bluriness.
- **3D Shape Realism:** That is the video in which main object has the most natural shape, again pay attention to deformed limbs of human and animals and unnatural deformations.
- **General Realism:** Please provide your subjective appraisal of which video, in your opinion, stands out as superior based on appearance quality, 3D structure quality, and motion quality.
- **Significance of Motion:** The video that contain the most amount of motion. Please exclude from consideration any random limbs deformations.
- **Video-text Alignment:** That is which video reflect all the aspects included in the text description.

You are shown a description of a video and two different videos generated by AI based on this description. Your task is to answer 7 questions regarding quality of these videos. Please pay close attention to instructions and answer as thoughfully as you can. In each task one question is a test one on which acceptance of the task will be judged. Please exclude size of the video from any considerations and pay attention only to the video content.

1) Which video has more realistic motion? Take into consideration of magnitude, smoothness, consistency of the motion. Pay close attention to deformed limbs of human and animals and unnatural deformations.

2) Which video has the most photo-realistic foreground? Assess realism of the appearance of the foreground object, do not consider the motion factor. That is the main object that looks more like taken with real video camera, pay attention to unnatural color or styles of the objects.

3) Which video has the most photo-realistic background? Assess realism of the appearance of the background. That is the background that looks more like taken with real video camera, pay attention to unnatural color or styles of the objects or bluriness.

4) Which video has object of better more realist shape? That is the video in which main object has the most natural shape, aggain pay attention to deformed limbs of human and animals and unnatural deformations.

5) In general which video looks more realistic?

6) Which video is most dynamic? The video that contain the most amount of motion. Please exclude from consideration any random limbs deformations.

7) Which video is better following text description? That is which video reflect all the aspects included in the text description

| **Text Description:** a humanoid robot playing the violin |
|---|

Video A                                                    Video B

**Which video has more realistic motion?**

○  Option A

○  Option B

**Which video has the most photo-realistic foreground?**

○  Option A

○  Option B

**Which video has the most photo-realistic background?**

○  Option A

○  Option B

**Which video has object of better more realist shape?**

○  Option A

○  Option B

**In general which video looks more realistic?**

○  Option A

○  Option B

**Which video is most dynamic?**

○  Option A

○  Option B

**Which video is better following text description?**

○  Option A

○  Option B

Figure 10: Screenshot of the user study webpage.

