# OpenReview forum: "4Real: Towards Photorealistic 4D Scene Generation via Video Diffusion Models"
_NeurIPS.cc/2024/Conference — NeurIPS 2024 poster_

### Official Review · Reviewer_vJ8m · 2024-07-05

**Soundness:** 3
**Presentation:** 2
**Contribution:** 3
**Rating:** 6
**Confidence:** 4

**Summary:**

4Real presents a novel framework for dynamic 3D scene generation. Given an input video, the framework selects a frame and prompts a text-to-video diffusion model to generate a "freeze-time video" that reconstructs the canonical 3D scene. Then, video-SDS-based optimization is used to optimize a dynamic 3DGS with reference to the input video. Compared to existing approaches that focus more on object-centric generation, the new method is able to generate more complex 4D scenes including multiple objects.

**Strengths:**

1. 4Real generates photo-realistic 180-degree 4D scenes for the first time. In comparison, previous 4D generation works are mainly object-centric, restricted by the object-centric image models that they rely on. 4Real only relies on native text-to-video diffusion models, thus addressing the bias. The approach is more favourable than 4Dfy and Dream-in-4D as suggested by a user study.
2. Although the work mainly follows the previous video-SDS-based 4D generation frameworks, it proposes several effective techniques to enable modeling more complex scenes that include multiple objects. A key design is to generate the canonical 3D through freeze-time video and video extension. An additional dynamic 3DGS fitting and a heuristic loss (small motion regularization) are proposed to offset the remained motions in the freeze-time videos, which I find reasonable and novel.
3. The paper also provides empirical values to the video-SDS-based 4D generation frameworks. It reduces the video-sds computational cost dramatically by leveraging low-resolution pixel diffusion models.

**Weaknesses:**

1. The description of context conditioning (L149) is not clear to me. "One dataset consists of static real-world objects with circular camera trajectories", is this dataset similar to [CO3D](https://ai.meta.com/datasets/co3d-dataset/), does it contain background, and how critical is the context embedding? Including this information can be important to reproduce this work.
2. The evaluation did not compare with AYG [28], whose results could also be easily accessed. AYG optimizes a dynamic 3DGS while the compared 4Dfy and Dream-in-4D optimize a dynamic NeRF, so I would say AYG is closer to this work. Although similar conclusions could be drawn for both AYG and 4Dfy/Dream-in-4D as they are all object-centric, the argument in L271 that claims "These methods are known to achieve some of the best visual quality compared to AYG and ..." is neither well-grounded nor convincing. L270 is also wrong to claim that Dream-in-4D is based on D-3DGS.
3. Generating a freeze-time video for the static 3D generation makes a lot of sense, but I wonder if it is possible that the generated freeze-time video could conflict with the reference video. Moreover, the canonical 3D generation stage seems to be solving a standalone image/text-to-scene task, it would be interesting to know if recent 3D scene generation works like RealmDreamer[A] and CAT3D[B] could be a drop-in replacement.

Minor:
- Editing is mentioned in the contributions, but it has not been discussed in any other places in the paper.

[A] Shriram et al., RealmDreamer: Text-Driven 3D Scene Generation with Inpainting and Depth Diffusion

[B] Gao et al., CAT3D: Create Anything in 3D with Multi-View Diffusion Models

**Questions:**

Please kindly address the weaknesses above.

**Limitations:**

Yes.

---

> ### Author Rebuttal · Authors · 2024-08-07
>
> **Context embedding.** First, the dataset for “static real-world objects with circular camera trajectories”, consists of data similar to the MV-ImagNet, with real-world backgrounds. Second, the use of context embedding is critical, as it controls the video model to generate freeze-time-like video. Without this step, the model predominantly generates dynamic videos whenever nonrigid subjects are included in the text prompt, regardless how the text prompt is engineered. The usage of context embedding can definitely be abandoned in the future when the video models are equipped with better camera-control and better consistency to input prompts.
>
>
> **Compare with AYG and correct description of Dream-in-4D.** We thank the reviewer for pointing out our mistake in wrongly describing Dream-in-4D as a 3D GS-based method. We will correct this error accordingly. Additionally, we have included a comparison against AYG, as shown in Figure 3 and Table 1 in the PDF. We find that our overall assessment of the advantages of our method over prior object-centric methods remains valid.
>
> **Consistency between reference video and freeze-time-video.** To minimize inconsistency, we generate reference videos without camera motion. This approach reduces disoccluded regions caused by camera movement, which is a major source of potential inconsistency between the reference video and the freeze-time video. Given this setup, we find the low-res video model is able to generate mostly consistent pixels in the remaining disoccluded regions caused by object motion, partially due to any inconsistency is hard to notice in low-res video. To further improve consistency, we merge two videos into a single video when feeding it into the video upsampler, leveraging the temporal self-attention in the upsampler to improve consistency.
>
> **Applying recent 3D scene generation works.** Yes, we believe recent image-mv models trained for 3d scene generation can be a drop-in replacement for the part of generating freeze-time video. Unfortunately, none of these concurrent approaches (i.e., RealmDreamer and CAT3D) are open-sourced yet, so it is not clear how they perform compared to the video model we employed. We will include a discussion of these methods in the revised version.
>
> **Video editing.** Thanks for the reminder. We mentioned the flexibility in selecting and editing videos in our contributions based on our generation-via-reconstruction pipeline that allows users to generate, select, and edit the reference video they want. We will include our results of applying simple video editing techniques, such as face-swapping or attributes manipulation, in the revised draft of the supplementary material.

---

> > ### Comment · Reviewer_vJ8m · 2024-08-11
> >
> > Thanks for the rebuttal. It clarifies most of my concerns. The numerical comparison with AYG looks good to me. Figure 2,3,4 in the rebuttal would be more helpful if more frames could be visualized.
> >
> > On 3D-video inconsistency, a static camera in reference video does not avoid the inconsistency as the objects can move as well, like turning around. "Merge two videos into a single video when upsampling" sounds like a reasonable trick, and should be discussed more in detail in the paper. It would also be better if more principled solutions could be discussed for future works.
> >
> > I will keep my score.

---

### Official Review · Reviewer_oa6s · 2024-07-11

**Soundness:** 3
**Presentation:** 3
**Contribution:** 3
**Rating:** 6
**Confidence:** 5

**Summary:**

The paper introduces a pipeline designed for photorealistic text-to-4D scene generation, discarding the dependency on multi-view generative models and instead fully utilizing video generative models trained on diverse real-world datasets. The method begins by generating a reference video using the video generation model, then learns the canonical 3D representation of the video using a freeze-time video, delicately generated from the reference video. To handle inconsistencies in the freeze-time video, the method jointly learns a per-frame deformation to model these imperfections, then learns the temporal deformation based on the canonical representation to capture dynamic interactions in the reference video.

**Strengths:**

1. The paper delivers impressive visual results for 4D scene level generation. The quality is outstanding and scene level generation is more general than previous object level generation.

2. The paper is well written with elaborated technical details.

**Weaknesses:**

1. The method seems extremely slow due to multiple steps being applied. The method hasn't provide enough latency comparisons.

2. Besides user studies, I wonder if any qualitative results can be shown, for example, FVD. Or some video sequences from multiview data (in door or outdoor) and compute reconstruction loss, but applying the lifting for reference video (using one view video in the data set) and render the other view to compare with the gt novel view video.

3. The model has a freeze time view deformation field, which indicates fitting static 3dgs scene from freeze-time video is unable to deliver good initial 3dgs field. How to balance the 3d consistency and plausibility from freeze time camera prior? More reasoning and analysis is desirable.

4. The paper uses freeze time SDS some part and multi-view SDS somewhere else. Better to improve the consistency.

**Questions:**

1. The paper mentioned a freeze time view deformation field. Does every frame has it or it is only applied to the first frame.

2. How important is the freeze time view deformation field.

3.  The motion video diffusion needs to sample with the first frame, how does the video diffusion handle different sampled fps? Using it as a input? This setting potentially limits the extension of this method to longer video.

**Limitations:**

Multiview video can be a better prior, it is limited by training data, but will improve in the future. 4D generation might be benefited from this prior and reduces the requirement of multiple stages.

---

> ### Author Rebuttal · Authors · 2024-08-07
>
> **Latency comparison.** In Ln. 64, we mentioned that our overall runtime is approximately 1.5 hours on a single A100 GPU, compared to over 10 hours for 4Dfy and Dream-in-4D. To provide additional latency analysis, we break down our runtime as follows: approximately 10 minutes for reference and freeze-time video generation, about 2 minutes for running COLMAP, around 20 minutes for reconstructing the canonical GS, and less than 1 hour for the remaining reconstruction steps with SDS.
>
> **Additional qualitative metrics.** We included additional metrics to quantitatively evaluate the rendered videos (see Table 1 of the PDF). We employ the XCLIP score [1], a minimal extension of the CLIP score for videos. We also run VideoScore [2], a video quality evaluation model trained using human feedback, which provides five scores assessing visual quality, temporal consistency, text-video alignment, and factual consistency. Our method significantly outperformed other methods in these metrics. However, we remain conservative about the effectiveness of these metrics since they have not been thoroughly studied for text-to-3D/4D generation.
>
>  Method            | **X-CLIP $\uparrow$** | **Visual Quality $\uparrow$** | **Temporal Consistency$\uparrow$** | **Dynamic Degree $\uparrow$** | **T-V Alignment $\uparrow$** | **Factual Consistency $\uparrow$**
> -----------------|-----------------------|-------------------------------|------------------------------------|-------------------------------|------------------------------|------------------------------------
>  **4Dfy**        | 20.03                 | 1.43                          | 1.49                               | 3.05                          | 2.26                         | 1.30
>  **Ours**        | **24.23**       | **2.43**               | **2.17**                      | **3.15**                 | **2.91**               | **2.49**
>  **Dream-in-4D** | 19.52                 | 1.34                          | 1.37                               | 3.02                          | 2.27                         | 1.20
>  **Ours**        | **24.77**        | **2.41**                 | **2.15**                      | **3.14**                 | **2.89**                | **2.46**
>  **AYG**         | 19.87                 | **2.49**                 | 2.09                               | 3.15                          | 2.80                         | 2.47
>  **Ours**        | **23.09**        | 2.44                          | **2.16**                      | **3.16**                 | **2.90**               | **2.50**
>
>
> Additionally, we excluded the FVD metric, as computing it with a limited number of rendered videos is statistically not meaningful. Evaluating with real-world multi-view videos is also not easily applicable to our pipeline. This is because we made specific design choices, such as requiring the reference video to have no camera motion to avoid inconsistency between the reference video and the freeze-time video. This assumption does not hold for casually captured dynamic videos.
>
>
>  **Analysis of freeze time view deformation field.** The reason for introducing view-dependent deformation (referred as per-view deformation in the paper) is to treat inconsistency in the generated freeze-time videos as geometric deformation. Without the view-dependent deformation, the canonical GS tends to underfit the freeze-time video, leading to blurrier regions, particularly in the background, where the video model often produces geometrically incorrect results.
>
> However, while view-dependent deformation can improve underfitting, it can also lead to overfitting to the reference video, resulting in noisy reconstruction. To mitigate this, we employ a regularization loss of the magnitude of the deformation (as mentioned in lines 180-186)  to prevent overfitting. Through empirical testing with a few samples, a weighting of 0.01 was found to be effective and is used consistently throughout the experiments.
>
> Additional visual comparisons are provided in Figure 5 of the PDF to support this analysis. These comparisons demonstrate the effects of removing either the per-view deformation or the small motion regularization.
>
> **Does freeze time view deformation field apply to other temporal frames?** No, it is only applied to fit the frames from the freeze-time video. It is not used for any other frames.
>
>
> **How does the video model handle different sampled fps?** The video model we used can take FPS as input. Additionally, as described in Ln. 142, it is a frame-conditioned model capable of performing either autoregressive generation or frame interpolation. Therefore, theoretically, our method can handle longer videos in the cost of longer processing time in the following reconstruction stage.
>
> **Multiview video prior.**  Yes, we also strongly believe that developing a generalizable multiview video diffusion model is one of the most promising directions for 4D generation. We briefly mentioned potential implementations, such as cross-frame attention, in Ln. 308. In the new draft, we will include a discussion of more concurrent efforts in multiview video generation. At this point, however, our method serves as a pioneering effort in demonstrating the possibility of generating realistic 4D scenes.
>
> **References:**
>
> [1]  Expanding Language-Image Pretrained Models for General Video Recognition.
>
> [2] VideoScore: Building Automatic Metrics to Simulate Fine-grained Human Feedback for Video Generation.

---

### Official Review · Reviewer_7sxF · 2024-07-12

**Soundness:** 2
**Presentation:** 2
**Contribution:** 2
**Rating:** 5
**Confidence:** 4

**Summary:**

This paper proposes a straightforward method to distill a 4D dynamic scene from a video diffusion model. Given a reference video (only generated videos are shown in the paper, no real video), they first generate a freeze-time video as a multi-view to reconstruct a reference 3DGS. Because they don't have a good multiview image model to capture the general scene multiview prior, they can only use a video model, which might produce small deformations. Some engineering effort is paid here to accept such deformation. After the reference Gaussians is reconstructed, an SDS stage is applied to both the frozen time multiview images and the temporal deformed frames.

**Strengths:**

- This is an early attempt to generate 4D scenes from video diffusion models
- Although very straightforward, the system works for generating some 4D videos.

**Weaknesses:**

- Object-centric? One important constraint this paper assumes is that because they want to generate dynamic scenes, they don't use sophisticated object-centric models. However, most video examples this paper shows do not have significant background motion, so one strong baseline/alternative is to just distill a static background and put a dynamic subject (from a dynamic object-centric SDS) in the foreground. Maybe this would achieve much better results than the paper's pipeline.
- How consistent? One question arises when you have a reference temporal video. Because there is a hallucination of the freeze-time view only considering a specific frame, however, the hallucinated freeze-time multiview images may not be consistent with what is observed in the temporal video, how does the system handle such in-consistency?
- Failure case: when i open the static website files in the supply, there is a folder called "good results", which the reviewer guesses may be because of some case selection, it would be critical to also show failure case.

**Questions:**

The authors also have to highlight their novelty, since the method is straightforward.

**Limitations:**

Some limitation is discussed, but no failure cases are shown.

---

> ### Author Rebuttal · Authors · 2024-08-07
>
> **Baseline by 3D scene generation + 4D object-centric generation.**
> Although this is a straightforward baseline, implementing it with high quality is challenging.
>
> - First, inserting 3D objects with realistic and physically correct placement is not trivial. Common issues include floating objects, misaligned scales between objects and background, and unrealistic scene layout.
>
> - Second, generating object motions that align well with the backgrounds is difficult, such as ensuring a generated target subject sits on a sofa rather than stands on it.
>
> - Third, complex procedures are needed to relight objects to match environment lighting.
>
> There exists several recent works either specialized in human-environment interaction [3] or generating simple 3D scene layout [4]. Currently, we are not aware of methods able to achieve all above points automatically.  We tried our best to create a few baseline results with heavy manual efforts, as shown in Fig. 2 in the PDF. We create the 3D background by removing the object from our generated freeze-time video. We then employ 4Dfy to generate the foreground objects. Finally, we manually insert the objects into the background with the most plausible position and scale we can find. Despite these efforts, we find the inserted object appears disconnected from the background, especially when compared to 4real’s results. For example, in the first example, the fox stands on the sofa instead of lying on the sofa; in the 2nd example, the racoon appears floating above the DJ equipment, and the lighting is inconsistent with the purple/blue environment lighting; and in the last example, the racoon appears popping out of the newspaper instead of lying on it.
>
> Overall, we think that 3D scene generation + 4D object-centric generation could be a viable alternative. However, this approach requires significant and complex research and engineering efforts. At this point, directly lifting 3D from generated videos, as demonstrated in our pipeline, appears to be a more generalizable and straightforward method. This is because the video model inherently produces realistic layouts, motion, and lighting. Furthermore, we envision that incorporating concurrent advancements in 4D reconstruction (e.g., MoSca [1]) and 4D object-centric generative priors (e.g., SV4D [2]) will further improve the quality of our pipeline.
>
>
> **Consistency between reference video and freeze-time-video.** To minimize inconsistency, we generate reference videos without camera motion. This approach reduces disoccluded regions caused by camera movement, which is a major source of potential inconsistency between the reference video and the freeze-time video. Given this setup, we find the low-res video model is able to generate mostly consistent pixels in the remaining disoccluded regions caused by object motion, partially due to any inconsistency is hard to notice in low-res video. To further improve consistency, we merge two videos into a single video when feeding it into the video upsampler, leveraging the temporal self-attention in the upsampler to improve consistency.
>
> **Failure cases.** As stated in Ln. 303, our method may fail under conditions such as rapid movements, the sudden appearance and disappearance of objects, and abrupt lighting changes. We will include failure samples in the new draft. To clarify what we mean by “good results,” we initially generated approximately one hundred results with various styles of reference videos without human filtering to understand the limits and ideal use cases of our method. After analyzing these results, we identified our limitations as mentioned above and retained the set of “good” videos—those with moderate movements, no sudden appearance or disappearance of objects, and no significant lighting changes—as example results for the intended use case of our method. The selection process can be entirely automatic via heuristic rules based on our observations.
>
> **Highlight of Novelty.** In the 3D generation field, the paradigm where a 3D reconstruction following multiview image generation is one of the mainstream approaches. Although the idea is straightforward, achieving convergence to this solution and making it work effectively is not trivial. Similarly, the proposed generation-via-reconstruction pipeline for 4D generation is non-trivial in practice. Naively combining existing video models and 4D reconstruction techniques does not yield desirable results. The major contributions and novelties of our approach are (1) a workflow to obtain plausible freeze-time videos from reference videos, given current video models of limited capability, (2) a deformation field to handle the inconsistencies in imperfect freeze-time videos, and (3) a joint temporal and multiview SDS to help reconstruct temporal deformation. With these proposed techniques along with other efforts to adopt various existing components and regularization effectively, we achieve the presented realistic 4D generation results.
>
>
> **Reference:**
>
> [1] MoSca: Dynamic Gaussian Fusion from Casual Videos via 4D Motion Scaffolds.
>
> [2] SV4D: Dynamic 3D Content Generation with Multi-Frame and Multi-View Consistency.
>
> [3] GenZI: Zero-Shot 3D Human-Scene Interaction Generation
>
> [4] GALA3D: Towards Text-to-3D Complex Scene Generation via Layout-guided Generative Gaussian Splatting

---

> > ### Comment · Reviewer_7sxF · 2024-08-07
> > **Keep my positive recommendation**
> >
> > After reading the reviews and rebuttals, I still think this paper provide valuable contribution to the community. The authors should include the above references [1-5] and discussions in the revision to better clarify the potential concerns, also, if possible, report the failure ratio that how many bad or good results will be generated from the 100 videos. I keep my positive recommendation.

---

### Official Review · Reviewer_yWZo · 2024-07-13

**Soundness:** 2
**Presentation:** 3
**Contribution:** 2
**Rating:** 6
**Confidence:** 4

**Summary:**

This paper proposes a new pipeline called 4Real, aiming for more photorealistic text-to-4D generation than prior work. The method first generates a reference video, then learn a canonical 3D representation from a freeze-time video. Afterwards, per-frame and temporal deformation are learned to model the gap between the canonical representation and the targeted video. This paper compares with two baselines and includes both visualizations on 30 examples and user study. The method is also much faster than prior work.

**Strengths:**

1. This paper has lifted the requirement of training a 3D generative model using limited synthetic data from prior work. This is important since there are much less available 3D assets than videos in the real world. This paper shows a potential way of fully unleashing the power of the abundant diverse video data in improving the text-to-4D models.

2. The model takes 1.5 hour for testing, which is much faster than 10+ hours from previous methods. Although it is still very slow compared  to other text-to-X models, it's already a great improvement.

3. The paper has a good presentation and is well-written. It also shows all the testing results in the supplementary.

**Weaknesses:**

This paper has clearly identified quite a few key limitations of prior work and aims to solve them, which is great. But I don't think all of these claims have been validated. For example:

(1) this work captures the interaction between objects and environments

From all the submitted video results, they do not seem to include much (or any) interaction between the object and the environment (e.g., relative global movement between them). For example, for the "bear driving a car" comparison between ours and 4Dfy, the car in "our result" is not moving on the lawn. The movement only comes from the bear inside the car.

(2) prior work being object-centric while this work not.

Although it's clear that the compared baselines only generate foreground object while the proposed method also creates the background environment. Since there's not much interaction between the background and foreground (as explained above), a better baseline would be generating an empty background/environment that suits the prompt/foreground and then putting the baseline-generated object properly in the environment.

**Questions:**

1. From the results, it does seem that the baselines generate much less realistic objects than the proposed method as the paper has claimed. However, existing text-to-3D models have proved their power in generating realistic 3D assets [20, 24, etc.], so could it because the 3D generation model used in baselines are a bit outdated or the prompts were not the best ones? Maybe the realism in the baseline results can be largely improved by a bit prompt engineering (e.g., adding keywords like "realistic" in the prompt) or updating their used 3D generation model?

2. Does the deformation model also time-varying appearance or only time-varying geometry?

3. Minor suggestions:

(1) maybe adding the input prompt to Fig. 4 may help the readers better understand the output content and quality.

(2) Some typos:

L24: ", E" -> ". E"

L292: missing a period.

**Limitations:**

The limitations seem to have been sufficiently discussed.

---

> ### Author Rebuttal · Authors · 2024-08-07
>
> **Interaction between objects and environments.** Our approach aims for a generalizable approach to generate realistic interaction between objects and environments, which includes natural multi-object placement, realistic environmental lighting effects, and relative motions between objects and environment. We acknowledge the ambiguity of the term “interaction” and will clarify these points. As per the reviewer’s request, we provide additional samples (see Fig. 4 in PDF) with moderate global motion between the object and the environment, such as “a storm trooper walking forward …” and “a dog running from left to right in front of a chair”. While our method currently faces challenges in reconstructing content with more rapid motion (as stated in Ln. 303), this limitation may be alleviated using concurrent advancement in 4D reconstruction method with long-term point tracking, such as MoSca [1].
>
> **Baseline by 3D scene generation + 4D object-centric generation.** Although this is a straightforward baseline, implementing it with high quality is challenging.
>
> - First, inserting 3D objects with realistic and physically correct placement is not trivial. Common issues include floating objects, misaligned scales between objects and background, and unrealistic scene layout.
>
> - Second, generating object motions that align well with the backgrounds is difficult, such as ensuring a generated target subject sits on a sofa rather than stands on it.
>
> - Third, complex procedures are needed to relight objects to match environment lighting.
>
> There exists several recent works either specialized in human-environment interaction [4] or generating simple 3D scene layouts [5]. Currently, we are not aware of methods able to achieve all above points automatically.  We tried our best to create a few baseline results with heavy manual efforts, as shown in Fig. 2 in the PDF. We create the 3D background by removing the object from our generated freeze-time video. We then employ 4Dfy to generate the foreground objects. Finally, we manually insert the objects into the background with the most plausible position and scale we can find. Despite these efforts, we find the inserted object appears disconnected from the background, especially when compared to 4real’s results. For example, in the first example, the fox stands on the sofa instead of lying on the sofa; in the 2nd example, the racoon appears floating above the DJ equipment, and the lighting is inconsistent with the purple/blue environment lighting; and in the last example, the racoon appears popping out of the newspaper instead of lying on it.
>
> Overall, we think that 3D scene generation + 4D object-centric generation could be a viable alternative. However, this approach requires significant and complex research and engineering efforts. At this point, directly lifting 3D from generated videos, as demonstrated in our pipeline, appears to be a more generalizable and straightforward method. This is because the video model inherently produces realistic layouts, motion, and lighting.
>
> **Could up-to-date text-3D models improve realism.** The  reason why 4Dfy and dream-in-4d is less realistic is because relying on text-MV model (i.e. MVdream) trained with synthetic data. As a result, the model is biased to generate synthetic-style objects, even when “realistic style” is added to the text prompt (see Fig.1). Some text-3D methods achieve a certain level of realism by using generative models trained with real data, either performing SDS with text-image model [2], or training image-MV model using real data [3]. In this regard, our approach can be loosely considered an updated version of these methods with an improved 3D prior.
> We will include this discussion in the next draft.
>
>
> **Does the deformation model also time-varying appearance?** No, we only model geometry deformation. We find that incorporating time-varying appearance worsens reconstruction results by making the 4D reconstruction problem more ambiguous.
>
> We thank the reviewer for the valuable comments, and will modify accordingly.
>
>
> **Reference:**
>
> [1] MoSca: Dynamic Gaussian Fusion from Casual Videos via 4D Motion Scaffolds.
>
> [2] HiFA: High-fidelity Text-to-3D Generation with Advanced Diffusion Guidance
>
> [3] CAT3D: Create Anything in 3D with Multi-View Diffusion Models
>
> [4] GenZI: Zero-Shot 3D Human-Scene Interaction Generation
>
> [5] GALA3D: Towards Text-to-3D Complex Scene Generation via Layout-guided Generative Gaussian Splatting

---

> > ### Comment · Reviewer_yWZo · 2024-08-13
> >
> > I have read through all reviewers' comments and the rebuttals by authors. My concerns have been addressed. Therefore, I would like to raise my rating to weak accept. Please add the relavant discussions in the rebuttal to the final version of the paper if accepted.

---

### Official Review · Reviewer_3AzP · 2024-07-14

**Soundness:** 4
**Presentation:** 4
**Contribution:** 4
**Rating:** 8
**Confidence:** 4

**Summary:**

The paper presents a method for 3D scene generation using video diffusion models. The main contribution is to present a method that does not require a multi-view generative model trained on synthetic data. The paper identifies the use of multi-view models as a strong limitation limiting existing works to generate high-quality scenes. The method first generates a video from the text prompt, and then also generates a second video starting from a reference frame that corresponds to a time-freezed camera scan of the scene. The second video allows for a static 3D reconstruction of the reference frame, that is then used in reconstructing the motion in the original video. The proposed method is novel and achieves better results than the state of the art.

**Strengths:**

The paper is very well-motivated and well-written. The key limitation in existing works is very clearly and explicitly identified.
The method is explained clearly. The paper does a good job explaining a complex pipeline and the supplemental provides further details.
The evaluations demonstrate very clear improvements over the state of the art.

**Weaknesses:**

The paper shows high-quality results that outperform existing work, and also present ablations. I was curious whether the identification of the use of synthetic data could also be demonstrated as an ablation, e.g., finetuning the snap video model on synthetic data and adding another SDS loss to demonstrate that it is indeed the key factor that limits high quality.

The paper in the intro mentions "reliance on multi-view generative models". While the video generative model used in the paper does not take camera poses as conditioning, it is in fact trained on static multi-view data. If so, I would still call such a model capable of generating multi-view images as a multi-view model. How would the overall approach perform if the snap video model was not trained on static multi-view scans? It would be good to provide more details on the training data. What percentage of the training data consists of such multi-view recordings?

The paper claims efficiency as a contribution; however, later in L225-228, it correctly points out that most of this improvement is due to the use of snap video model, and not due to any contributions of this paper. If true, the claims should be adjusted.

From L161-163, it seems like the sufficient amount of pose variation is chosen manually? Is it not possible to automate this? If this step is manual, how is it accounted for in the reported processing times?

**Questions:**

Does Colmap work for all the dynamic videos generated, even if the motion is large?

The results are generally very impressive but it would be very helpful to the community to also show failure examples so that the remaining challenges are clear.

**Limitations:**

Yes.

---

> ### Author Rebuttal · Authors · 2024-08-07
>
> **Ablation of finetuning with synthetic data.** This is a great suggestion to rigorously prove the benefit of using models trained with real world data. However, conducting such experiments requires significant GPU hours to reach a meaningful conclusion, which we are unable to fulfill for rebuttal. Instead, we refer to results from text-MV image models finetuned using synthetic data, e.g. MVDreamer. As shown in Fig 1 of the PDF, images generated by MVDreamer are cartoonish, even when “realistic style” is included in the input text prompt.
>
> **Importance of training with static multi-view data.** Training video models with static multi-view data is crucial not only because it supplements the training set with videos of static scenes but, more importantly, because it allows us to learn a context embedding (associated to the multi-view dataset) that serves as a tool to control the model to generate freeze-time-like video. Without this step, the model predominantly generates dynamic videos whenever nonrigid subjects are included in the text prompt, regardless how the text prompt is engineered.
>
> **Details of training data.** The full training set combines a dozen of datasets, each corresponding to different type/style of videos, and is associated with a context embedding. For the static multi-view dataset, it is mainly created from MVImageNet, and sampled based on a predefined ratio during training. The ratio is set according to the number of video hours in each dataset, which results in roughly 1% for the static multi-view dataset.
>
> **Efficiency contribution claim.** It is partially correct that one source of efficiency comes from the video model we used. However, the main source of efficiency, as stated in Ln. 58, is that we "transform the generation problem into a reconstruction problem and reduce reliance on time-consuming score distillation sampling steps." Specifically, as described in Ln. 263, during each stage of the reconstruction procedure, we perform SDS only during the last 5,000 iterations out of the total 20,000 iterations. This choice significantly reduces runtime, as each SDS iteration takes over 20 times longer than simply minimizing reconstruction losses.
>
> **Is choosing amount of pose variation automatic?** This part is automated in our experiments. First, we perform auto-regressive view extension until the video reaches a maximum of 72 frames, which is often more than enough for 180+ degree coverage. Next we run Colmap with the generated video. If Colmap fails to converge, indicating the current video is significantly geometrically inconsistent, the video is automatically shortened until Colmap is able to reach a solution. We will include this in the supplementary.
>
> **Does Colmap work for all the dynamic videos generated?** We recognize the challenge of pose estimation for dynamic videos. Therefore, we purposely generate dynamic videos without camera motion. This is achieved by using text prompts “camera is static”. We will include this detail into the supplementary.
>
> **Failure case.** As stated in Ln. 303, our method may fail if there are “rapid movements, the sudden appearance and disappearance of objects, and abrupt lighting changes”. We will include failure samples in the new draft.

---

> > ### Comment · Reviewer_3AzP · 2024-08-11
> >
> > Thanks for the response. I would still argue for acceptance. I would urge to add more rigorous experiments to study the role of synthetic data in the final version if accepted. For efficiency, papers such as Wonder3D (and more recently Cat3D) demonstrate 3D results without score distillation. It would be good to tone down the arguments there and mention both the role of the snap video model and the existence of other non-SDS papers.

---

> > > ### Author Response · Authors · 2024-08-13
> > >
> > > Thanks for the great suggestions! We will add more rigorous analysis with synthetic data, and include thorough discussion of related concurrent works.

---

### Author Rebuttal · Authors · 2024-08-07

We would like to thank the reviewers for the positive feedback and insightful comments. We appreciate all the suggestions and will revise the paper accordingly.

To address major concerns from the reviewers, we include the following additional results in the PDF:

- **Additional baseline results from combining 3D scene generation and 4D object-centric generation (@yWZo, 7sxF)**: We found that this seemingly straightforward baseline is actually non-trivial to produce high-quality results with realistic object placement, object motion, and lighting effects, even with significant human intervention such as setting the placement position and object scale.
- **Additional results on text-mv models (@3AzP, yWZo)**: Text-MV model trained with synthetic data is biased to generate synthetic-style images, even when ”realistic style” is added to the text prompt.
- **Examples showing some global relative motion between the object and the background (@yWZo)**
- **Comparison against AYG (@vJ8m)**
- **Quantitative evaluation using XCLIP and VideoScore (@oa6s)**: XCLIP is a minimal extension of the CLIP score for videos. VideoScore is a video quality evaluation model trained using human feedback, which provides five scores assessing visual quality, temporal consistency, text-video alignment, and factual consistency.
- **Additional visual ablation of deformation (@oa6s)**: We add visual analysis of per-view deformation and the effect of using deformation regularization to balance between 3D consistency and plausibility from freeze time video prior.

We will elaborate on and address every question and suggestion in the following individual response.

---

### Decision · Program_Chairs · 2024-09-25

**Decision:**

Accept (poster)

**Comment:**

The paper proposed a text-to-4D generation framework, which first generates a canonical 3DGS from a generated freeze-time video and further optimizes with a video SDS-based loss to incorporate dynamics. The proposed pipeline enables generating more complex 4D scenes over video-SDS baselines (e.g. 4Dfy), overcomes the limitation on limited 3D/4D data, and provides promising results on 4D generation.

After the rebuttal, all the reviewers are positive about the paper. The AC agrees with the reviewers' suggestions on accepting this paper. As promised during the rebuttal, please add the additional comparisons with baselines, as well as additional results provided in the rebuttal  to support the claims mentioned in the paper.